# Surveys of Knowledge and Awareness of Plastic Pollution and Risk Reduction Behavior in the General Population: A Systematic Review

**DOI:** 10.3390/ijerph22020177

**Published:** 2025-01-27

**Authors:** Caterina Caminiti, Francesca Diodati, Matteo Puntoni, Denisa Balan, Giuseppe Maglietta

**Affiliations:** Clinical and Epidemiological Research Unit, University Hospital of Parma, 43126 Parma, Italy; ccaminiti@ao.pr.it (C.C.); fdiodati@ao.pr.it (F.D.); denisa.balan@ao.pr.it (D.B.); gmaglietta@ao.pr.it (G.M.)

**Keywords:** plastics, environment, pollution, knowledge, awareness, public health, questionnaire, survey, systematic review, cross-sectional

## Abstract

Individual attitudes and knowledge can predict pro-environmental behaviors. Public surveys, therefore, can provide precious information, which can guide sensitization interventions. In this systematic review, we searched Medline and Embase, with no language or date restrictions, for surveys designed to measure in the general population the level of knowledge about different types of plastics, the risks associated with plastic pollution, and awareness of actions to reduce them. Survey tools were analyzed following the guide of Burns and Kho, and study methodological quality was assessed via the Appraisal Tool for Cross-Sectional Studies. We included 17 articles published from 2019 to 2024, mostly concerning European populations. The tools comprised a median of 13 items (range 7–50), and very differently formulated questions. Overall, 13/17 (76.5%) study questionnaires received less than 50% (<3.5) of the maximum possible score. The remaining four questionnaires obtained intermediate scores (between 3.5 and 5.3) indicating moderate quality. Most studies did not employ the appropriate cross-sectional survey methodology, only two studies statistically justified sample sizes, only three reported a sampling frame, and only two described a selection process that appears to be representative. In most cases, the instruments were not validated, and the statistical significance of key variables was not provided. The many shortcomings highlighted in this review emphasize the urgent need for methodological rigor when conducting survey studies, which are essential tools for public health.

## 1. Introduction

The many useful properties of plastics, such as lightness, strength, resistance to corrosion, and low production costs, make them ideal for a wide range of applications, from agriculture to industry and technology. However, concern about the negative impacts of these materials on the environment and on human health is growing [1,2]. Plastic can harm the environment and human health in different ways, according to its physical and chemical characteristics [2,3,4] and to the phase of its lifecycle [5,6]. To protect human and planetary health from the increasing threats posed by plastics, international bodies such as the European Commission [6] and the United Nations [7] have called for immediate global action. In line with these recommendations, governments worldwide are enforcing plastic restrictions and plastic waste management rules. Nevertheless, plastic use is still growing exponentially, with global production expected to exceed 590 million tons in 2050 [8], indicating that no simple solution exists and that all societal stakeholders need to be involved [9]. This includes the active engagement of individual citizens, whose contribution is essential in promoting change and supporting effective environmental action [10].

The literature suggests that individual attitudes and knowledge about plastics can predict pro-environmental behaviors, and that exploring public opinion and knowledge is pivotal for the successful implementation of policies targeting plastic pollution [11]. Thus, to develop effective solutions to this evolving global challenge, it is essential to focus on public behavior, gathering insights into what citizens know about plastic and its risks [9,12]. Similarly, it is necessary to understand the level of public awareness of possible actions that people can take at an individual level to counter such risks [13]. To this end, surveys can provide valuable information that can constitute the basis for the development of public interventions, support policymaker decisions, and inspire future studies. With respect to plastic pollution, there is growing interest in understanding citizens’ levels of knowledge and awareness; however, uncertainty remains about the most suitable tools and methods to ensure data comparability across different contexts. To fill this gap, this systematic review aimed to identify, synthesize, and evaluate surveys designed to measure in the general population the level of knowledge about different types of plastics, the risks associated with plastic pollution, and the awareness of actions to reduce potential harm. The specific objectives were as follows:

-Identify validated and generalizable survey instruments;-Analyze the content of the questions used by these surveys, categorize them into themes, and highlight any shortcomings.

The findings of this study provide information on available tools and may help in the selection of the most appropriate instrument according to the desired purpose.

This work was conducted within the framework of a large, nationwide project financed by the Italian Ministry of Health [14].

## 2. Materials and Methods

Before initiating this work, in February 2024, we checked the PROSPERO database [15] for any ongoing review with the same study question, but none was found. The review was designed and conducted in accordance with the Preferred Reporting Items for Systematic reviews and Meta-Analyses (PRISMA) guidelines [16] (Appendix A). The protocol was registered with PROSPERO (CRD42024552230) on 29 May 2024.

### 2.1. Eligibility Criteria

#### 2.1.1. Study Eligibility

We included studies published in peer-reviewed journals that reported surveys aimed at recording the knowledge, awareness, and attitudes of the general population concerning plastic pollution, potential risks, and relative behavior. Survey tools could be administered in any format. We excluded (a) reviews, case–control studies, protocol studies, and pilot studies; (b) studies evaluating risk factors associated with plastics rather than awareness of those risks; (c) studies that described and/or evaluated the effectiveness of an intervention on knowledge and behavior; and (d) qualitative studies, such as focus groups, interviews, etc., that solely aimed to develop or validate questionnaires, or to design educational campaigns.

#### 2.1.2. Population Eligibility

As this review focused on the general population, we excluded research on specific groups (e.g., health professionals, subjects with specific diseases, children, students, or elderly individuals). An exception was made for surveys administered to participants with certain characteristics but using a questionnaire not specifically designed for that particular population subgroup.

### 2.2. Search Strategy and Literature Selection

An information specialist designed the search strategies and performed searches on Medline (PubMed platform) and Embase, with no language or date restrictions. The original search was conducted on 7 February 2024, and was rerun on 12 September 2024. A “backward” snowball search was conducted on the references of systematic reviews and relevant papers. The full search strategies for each database are provided in Appendix A.

Title and abstract screening was performed independently by two reviewers via the Rayyan platform [17], and discrepancies were resolved by discussion. The full texts of potentially eligible publications were reviewed by two reviewers independently, with disagreements again resolved by discussion. The reasons for exclusions were documented at this stage.

Two review authors extracted data from the included studies into a standardized, prepiloted form, and disagreements were resolved by consensus. The following data were collected: first author, year of publication, country, objective, population eligibility criteria, number of respondents/sample size, sampling technique, mode of administration, reference to tool development, and key instrument characteristics.

### 2.3. Thematic Areas

Two reviewers, in collaboration, classified each tool according to its content and aims, to verify whether the survey addressed any of the three thematic areas defined a priori in the protocol following the literature analysis, which indicated their relevance in determining citizen pro-environmental behavior.

#### 2.3.1. Level of Knowledge About Different Types of Plastics

“Plastics” is a generic term referring to a variety of materials with different physical and chemical characteristics. Furthermore, with increasing efforts to find substitutes for conventional plastics, terms such as biobased plastics, bioplastics, and biodegradable plastics are increasingly being used but with considerable ambiguity in the use of terminology [18].

#### 2.3.2. Level of Knowledge of the Risks Associated with Plastic Pollution

Plastics can potentially contribute to direct or indirect risks to human health through different mechanisms, depending on their characteristics [4]. The questionnaire should analyze the level of knowledge of the population regarding the environmental risks associated with a wide range of potentially toxic chemicals that can disperse in the environment and represent potential health hazards [4,5,6,19].

#### 2.3.3. Awareness of Actions to Reduce Potential Harm

As explained by theoretical frameworks [20], subjective factors such as attitudes, perceptions, motivation, etc., can influence individual pro-environmental behavior. Therefore, sensitization interventions targeting the public should also be grounded in an analysis of citizen awareness of available actions. In this review, we referred to the definition of awareness by Trevethan [21], which clarifies its distinction from knowledge. While knowledge indicates factual information acquired from authoritative sources, awareness has a strong personal element, including domains such as attentive self-perceptions about conditions related to health, apprehension about prospective health problems, strength of personal concerns, and awareness about one’s own need to engage in health-enhancing behavior [21].

### 2.4. Quality Assessment

#### 2.4.1. Quality Evaluation Tool

The analysis of the survey tools used in the included studies was performed by two reviewers independently, and disagreements were resolved through discussion, following the survey assessment guide by Burns and Kho [22]. The guide comprises seven main questions, and specific subquestions designed to help readers systematically appraise the quality of survey reports (Appendix A). Since the instrument was not designed to score the quality of studies, we implemented the scoring system devised by Tolonen et al. [23] for the same tool. Accordingly, the total quality of the included surveys was calculated based on the scores obtained from the main questions, with the maximum score being 7 points. Specifically, each main question comprised a minimum of two and a maximum of eight subquestions. The score for each main question was calculated as the sum of the scores for its subquestions. When all subquestions met the criteria, the main question was awarded a score of 1. Consequently, the weight of each subquestion varied depending on the total number of subquestions within each main question.

#### 2.4.2. Critical Appraisal of Study Quality

Two researchers independently assessed the methodological quality of the included studies, with disagreements solved through discussion, using the Appraisal Tool for Cross-Sectional Studies (AXIS) [24]. AXIS is a descriptive quality assessment tool comprising 20 components covering five main study sections: introduction, methods, results, discussion, and other. Since high-quality and complete reporting of studies is a prerequisite for judging quality, the instrument incorporates some quality of reporting as well as quality of design and risk of biases [24].

### 2.5. Statistical Analysis and Data Synthesis

Data were aggregated through narrative synthesis (year of publication, country, and inclusion and exclusion criteria), representativeness (sample size and sampling technique), theme of questionnaire (knowledge, awareness, and other), and validity and reliability of used questionnaires (if reported).

## 3. Results

### 3.1. Study Selection

The PRISMA flow diagram for the identification, screening, and inclusion of studies is provided in Figure 1.

The original search run on 7 February 2024 retrieved a total of 2153 records, which were uploaded into the Rayyan platform together with an additional article detected by the snowball search. After deduplication, 1572 records remained for manual title and abstract screening, of which 47 were considered potentially eligible and underwent full text review. Of these, 17 publications fulfilled the eligibility criteria and were included in the systematic review [25,26,27,28,29,30,31,32,33,34,35,36,37,38,39,40,41]. The two articles by Dilkes-Hoffman et al. [30,31] concerned research conducted with the same survey tool but reported results for two distinct sets of questions with different study objectives; therefore, we considered them two separate studies. The 30 excluded reports and the corresponding reasons for ineligibility are presented in Appendix A. The search was rerun on 15 September 2024, and an additional 145 deduplicated records were retrieved, none of which were selected for full text review.

### 3.2. Study Design and Participant Characteristics

Information on the seventeen included studies is provided in Table 1.

The years of publication ranged from 2019 to 2024. Most studies (numbering 10/17, 58.8%) exclusively concerned European populations, with two studies performed in Italy [26,34], two in Portugal [38,40], one in Greece [27], one in Germany [37], one in Poland [39], and three in multiple European countries [25,28,32]. Among the remaining studies, two were conducted in Australia [30,31], two in China [29,36], one in Spain and Mexico [35], and two in multiple countries across the globe [33,41]. The majority of the surveys (12/17, 70.6%) were self-administered electronically on internet platforms or by email [25,27,30,31,32,33,34,35,36,37,40,41]; one was self-administered in hard copy format [38], two employed face-to-face interviews [28,29], and one was a computer-aided interview [39]. For one study [26], the mode of self-administration was unclear. Three studies employed random sampling techniques [28,30,31], six used the snowball methodology [27,34,35,36,39,41], and seven used other nonrandom approaches [25,26,29,32,33,37,40], whereas Miguel et al. [38] did not indicate the sampling method. Ten papers [25,26,27,28,30,31,34,35,36,37] provided citations to studies used as references for questionnaire development, but most studies did not follow a formal methodology for the instrument validation process. Surveys had a median of 13 questions (Interquartile Range (IQR) 11–19) with a wide range of items from 7 to 50, and very differently formulated questions. Most surveys used responses on point Likert scales (ranging from 4 to 7 points) and/or multiple choice. In total, 37,643 participants (from 127 to 27,083 participants) were included in these studies, but representativeness was not ensured. In fact, only two studies defined a sample size [37,39], and four reported a response rate [28,29,30,31].

### 3.3. Main Thematic Areas

The classification of survey questions has proven difficult because of confusion in terminology (e.g., the words “knowledge” and “awareness” are often used interchangeably). Furthermore, only eight of the examined papers [29,30,31,33,35,36,37,41] included the questionnaire in its entirety; therefore, for the remaining articles, judgments had to be based on the manuscripts only. In Appendix A, we present a summary table showing for each study an assessment of whether the three predefined thematic areas were addressed. The most commonly explored of the three themes concerned knowledge on the risks associated with plastics, investigated by fourteen tools [25,26,27,30,31,32,33,34,35,37,38,39,40,41], followed by topics related to awareness of actions to reduce possible harm from plastic pollution, present in eleven tools [25,28,29,30,32,34,35,36,38,39,40]. Fewer instruments, nine in number, investigated knowledge about the different types of plastics [27,29,31,32,33,34,35,39,40].

Five questionnaires [32,34,35,39,40] comprised items concerning all three themes.

### 3.4. Assessment of Methodological Quality

Assessments of each survey report were performed following the guide by Burns and Kho [22] and applying the scoring in accordance with Tolonen et al. [23], as provided in Appendix A and depicted schematically in Figure 2.

Overall, 13/17 (76.5%) studies received fewer than 50% (<3.5) of the 7 points assigned when all the quality requirements were met. The remaining four surveys [30,31,35,38] obtained intermediate scores (between 3.5 and 5.3), which indicates the moderate quality of the instrument used. Quality was compromised, in particular, by a nonsystematic or only partially systematic approach in the development of the questionnaire (12/17 and 5/17, respectively); by the absence, in nearly all cases, of response rate data (15/17) and of strategies adopted to improve the response rate (8/17), which increases the risk of an unrepresentative sample and decreases the generalizability of the results; and by the complete or partial absence of a validation process for the measurement of psychometric properties (7/17 and 10/17, respectively). Specifically, in terms of the latter aspect, clinimetric testing was performed in only five studies (see Appendix A), most of which did not clearly report the number and type of participants involved or the results of the assessments.

The results of the methodological quality assessments performed with the AXIS tool [24] for each study are shown in Appendix A, and are represented graphically in Figure 3.

Our analysis revealed that 15/17 publications had clear study objectives focused on measuring levels of knowledge and awareness or attitudes and behaviors related to reducing risk factors associated with plastic pollution. Unfortunately, most did not employ the appropriate cross-sectional survey methodology consistent with the objectives. Furthermore, only two studies statistically justified the sample size [37,39]. With respect to the representativeness of the investigated samples, nine surveys defined the reference population [25,26,27,28,29,30,31,35,39], but only three of them [26,30,31] reported a sampling frame, and only two [30,31] described a selection process that appeared to be representative. Most studies used an instrument without testing its validity and without providing the statistical significance of key variables (reproducibility, reliability, etc.).

Concerning reporting of results, more than half of the studies presented adequate baseline data and almost all (except Menzel et al. [37]) reported results in accordance with the described methods. No study has addressed and categorized nonrespondents, or reported their main characteristics. All the publications except four [25,27,30,31] discussed limitations, including selection bias [26,32,33,34,35,37,38,39,40], measurement bias [26,28,34,35,36,37,38,39,40], limitations of the study design [26,29,34,41], and other possible confounding factors [38,41].

Concerning conflicts of interest and ethical reviews, five studies did not clearly declare funding sources that may have influenced the authors’ interpretation of the results [26,30,31,32,37]. Four studies stated that they did not request ethical approval or participant consent [25,27,32,33], while five did not mention these aspects at all [26,28,30,31,34].

## 4. Discussion

To our knowledge, this is the first attempt to systematically identify and analyze studies aiming to capture public knowledge and awareness of plastic pollution and related topics using public surveys. The broad search strategies we designed ensured that we obtained a comprehensive picture of current peer-reviewed research in this field.

The main finding of this work concerns the several critical shortcomings in the design and methodology of the included studies revealed by our careful evaluation of the quality of the instruments used and of the methods of conducting the studies.

All included studies provided a fairly clear definition of objectives, focused on assessing levels of knowledge, awareness, or attitudes and behaviors related to plastic use. To address these objectives, a cross-sectional survey design is appropriate, which aims to describe the prevalence of certain factors in the population at a given time. Unfortunately, as pointed out by our assessments using the AXIS tool (Figure 3), none of the studies were adequately designed and conducted, so the robustness of the results of these surveys was questionable. In particular, as sampling strategies to ensure representativeness of the target population were not implemented, the results were affected by selection bias. Although sample sizes are usually limited by the budget available for very large surveys, a representative sampling frame is essential for the generalization of survey results to the target population. Other problems we observed, applying the tool by Burns and Kho (Figure 2), concerned the survey instruments, which were often not developed following a rigorous process (including, for example, focus groups and expert panels for item generation, formatting, and sequential ordering), and were rarely validated. Finally, many of the included studies did not provide references for questionnaire design, suggesting limited use of the relevant literature. Notably, we cannot rule out that this information was not indicated in the papers because of incorrect reporting, which is also a common problem addressed by reporting guidelines.

The shortcomings described above are not surprising, as they have been extensively reported in the literature in other research areas [69,70]. In fact, analyses of survey quality in various disciplines have pointed to wide variations in survey design and inconsistencies in reporting practices, including failure to report essential metrics, such as response rates, and to provide complete questionnaires [71,72,73,74]. These limitations persist, despite the availability of guidelines that can be used as references, such as the Checklist for Reporting of Survey Studies (CROSS) [69] and the guide by Burns and Kho on how to assess a survey report [22].

This work has some limitations. Firstly, this review does not provide a quantitative synthesis of the study results because of the difficulty of combining questions phrased heterogeneously and using different modes of response collection (Likert scales, binary scales, and open questions). This was complicated by the fact that fewer than half of the examined papers included the entire questionnaire, and we did not succeed in obtaining the missing tools from the authors. Furthermore, the lack of original tools affected our attempt to classify survey questions according to their scope. This task was further complicated by the unclear or ambiguous phrasing found in many papers. Specifically, the distinction between “knowledge” and “awareness” was often blurred, with the two terms sometimes used interchangeably, a problem already reported in the literature [21].

## 5. Conclusions

Plastic pollution is an undeniable threat to health, and public surveys can be essential to guide the development of sensitization and education campaigns, but they must be undertaken with the same rigor applied to other clinical research studies. Surveys are known to be the most widely abused form of research because of their perceived ease of conduct. It is important to recognize the significant limitations of survey designs owing to their observational nature, and to ensure that the conclusions are justified and not “overstated”.

This systematic review fills a gap in the current literature by providing a synthesis and evaluation of existing surveys intended for the general population, measuring knowledge on different plastic types, risks associated with plastic pollution, and awareness of actions to reduce potential harm. Our analyses led us to conclude that no validated and generalizable survey instrument is currently available in this field, since most examined studies did not exhibit the requirements pertaining to survey reporting and methodological quality recommended in the literature. The Australian tool reported by Dilkes-Hoffman et al. appears to be the most compliant with such requirements; however, it requires further testing in other settings with rigorous studies.

## Figures and Tables

**Figure 1 ijerph-22-00177-f001:**
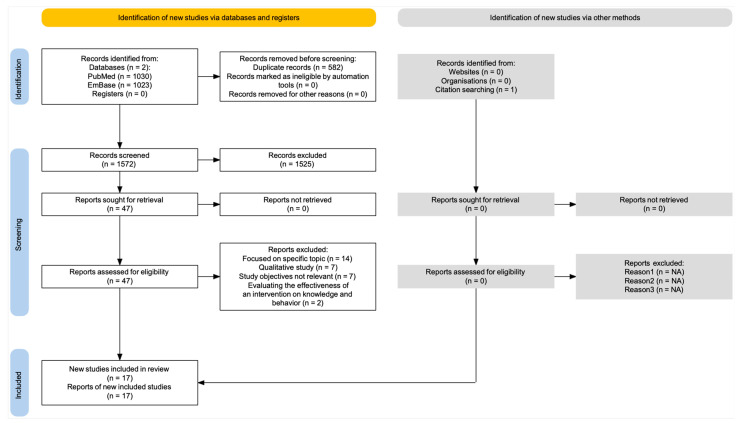
Prisma flow diagram.

**Figure 2 ijerph-22-00177-f002:**
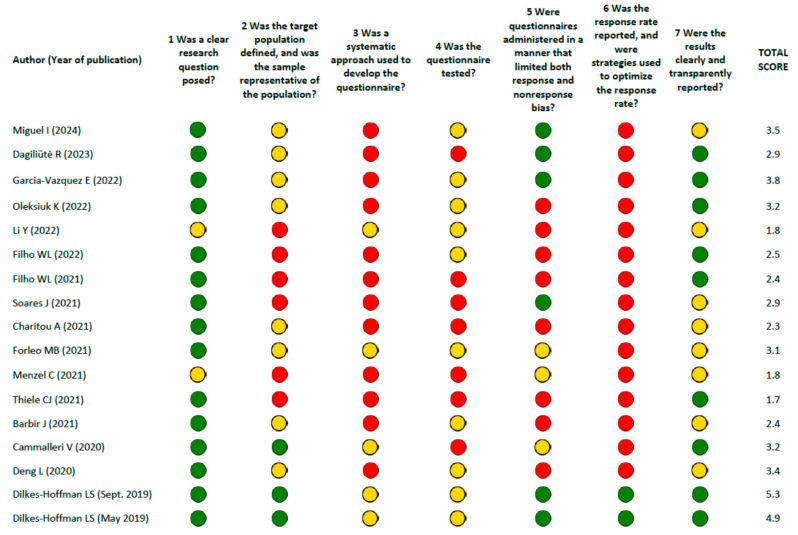
Methodological quality assessments of survey reports. Note: Each survey was evaluated by seven main questions and their subquestions. The sum of the points indicates the total quality of the survey (see the Quality Evaluation Tool subsection in the Section 2, and Appendix A). Colors were assigned to each main question based on the following criteria: green circle if the sum of subquestions answered “yes” is ≥0.66; yellow circle if the sum of subquestions answered “yes” is 0.33–0.65; red circle if the sum of subquestions answered “yes” is <0.33.

**Figure 3 ijerph-22-00177-f003:**
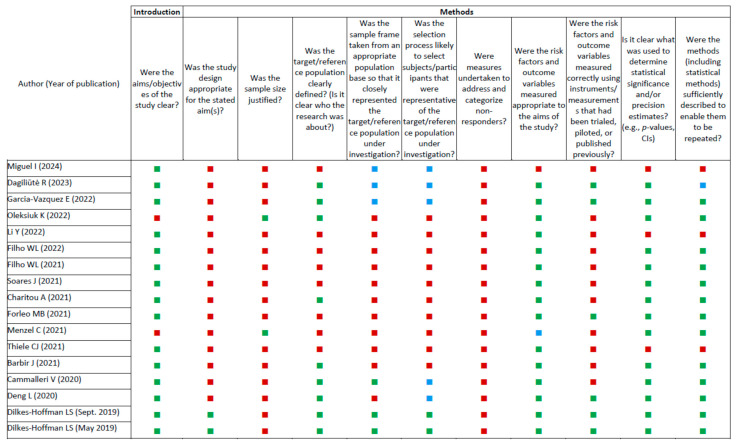
Quality assessment of the 17 included studies using the Appraisal Tool for Cross-Sectional Studies (AXIS). Note: to facilitate consultation, colors were used as follows: green = Yes, red = No, blue = Cannot tell.

**Table 1 ijerph-22-00177-t001:** Characteristics of the 17 included studies.

**Ref. #**	**First** **Author**	**Year of Publication**	**Objective of Study**	**Country**	**Inclusion Criteria**	**Exclusion Criteria**	**Number of Respondents/Sample Size**	**Sampling Technique**	**Administration**	**Reference of Tool Development**	**Key Instrument Characteristics**
[38]	Miguel I	2024	(i) To clarify the factors underlying environmental consciousness, concerns, and behaviors; (ii) to assess how participants’ sociodemographic characteristics affect these perceptions, in order to tailor more specific initiatives aimed at increasing environmental knowledge and encouraging pro-environmental behavior.	Portugal	General public (recruitment method is unclear and no specific criteria are indicated)	Not indicated	1129/NA	Not indicated	Self-administration (hard copy)	Not indicated	13 questions. Response collection: five-point Likert scale. To ensure face validity, the questionnaire was previously submitted to a pilot study to identify potential weaknesses of the questionnaire and test if the questions were formulated in a clear and understandable way. The feedback from the 5 subjects participating in the pilot study was discussed and considered for the final version.
[28]	Dagiliūt R	2023	To analyse the self-reported actions related to reduction of plastic and MP pollution by Europeans and factors influencing actions undertaken.	European Union	General public (recruitment method is unclear and no specific criteria are indicated)	Not indicated	27,083/27,498 (98.5%)	The Eurobarometer datasets provide for two types ofweighting, a post-stratification weighting and a population size weighting.	Face-to-face interviews	[42]	11 questions. Response collection: dichotomous responses; 4-point Likert scale. The study used the Eurobarometer survey on environmental attitudes, which is not intended for plastic pollution directly. The validity and reliability of scales was tested using Cronbach’s α. This coefficient for all scales ranges between 0.526 and 0.771, indicating moderate reliability of the scales.
[35]	Garcia-Vazquez E	2022	To determine the differences between Mexico and Spain in terms of behavior and behavioral intentions regarding MP.	Spain, Mexico	University students in Spain and Mexico	Not indicated	572/NA	Snowball methodology	Self-administration (online)	[29,43]	13 questions. Response collection: multiple choice; 7-point Likert scale. The questionnaire was developed based on previously published tools and examined by a panel of 6 experts. The Content Validity Index obtained from the expert panel was 0.96. The internal consistency was measured based on Cronbach’s α values (0.94 for items measuring MP risk awareness).
[39]	Oleksiuk K	2022	To research the current knowledge and awareness of consumers regarding sources, exposure, and health hazards connected to microplastics’ presence in water and foods, especially their impact on internal organs, metabolic processes, and reproductive functions.	Poland	In order to take part in the questionnaire survey, it was required to be aged 18 or above, live in Silesia (Poland), and be a student.	Not indicated	410/NA	Snowball methodology	Computer-aided web interview, via Google Forms	Not indicated	26 questions. Response collection: single and multiple choice. The questionnaire was validated for reliability, correctness, and relevance. Repeatability of the responses was examined by distributing the questionnaire twice to a random sample of 20 people. A total of 78.3% of the questions obtained very good agreement; Kappa > 0.75.
[36]	Li Y	2022	To determine whether there are gender differences in people’s pro-environmental psychology and behaviors in China.	China	No specific criteria are indicated. Recruitment of respondents was completed through social media platforms utilizing preexisting social and personal contacts.	Not indicated	532/NA	Snowball methodology	Self-administration (online)	[44,45,46]	7 questions: four kinds of pro-environmental psychology questions and three for pro-environmental behaviors. Response collection: 5-point Likert scale; dichotomous responses. The questionnaire was pilot tested on 25 respondents to revise the wording of the survey items so that the statements were appropriate.
[33]	Filho WL	2022	To explore the level of awareness and attitudes about bioplastics	42 countries located mostly in Europe and Asia	No specific criteria are indicated. Using the LimeSurvey platform, the questionnaire was made available electronically in several countries for five months.	Not indicated	384/NA	Using the LimeSurvey platform, the questionnaire was made available electronically in several countries for five months.	Self-administration (online)	Not indicated	19 questions. Response collection: multiple choice; 4-point Likert scale. Pre-testing by a subset of the respondents was carried out to identify and mitigate language and understanding problems.
[32]	Filho WL	2021	To investigate some of the main trends in plastic consumption, hence offering a better understanding of the effects of plastic pollution on the environment and the problems related to plastic use as perceived by consumers. Furthermore, the extent of current efforts on how to reduce plastic consumption was assessed. Special attention was paid to an analysis of the awareness of citizens of bioplastics, their usage, and environmental impacts.	16 European countries	No specific criteria are indicated. The survey was disseminated to all partners of the Horizon 2020 project BIO-PLASTICS EUROPE and in European JISCMail mailing lists related to sustainability and sustainable consumption.	Not indicated	127/NA	The survey was disseminated to all partners of the Horizon 2020 project BIO-PLASTICS EUROPE. Additionally, the survey was also disseminated in European JISCMail mailing lists related to sustainability and sustainable consumption. The link remained active during February and March of 2020 and received 127 responses from 16 European countries.	Self-administration (online)	Not indicated	18 questions. Response collection: 5-point Likert Scale. The questionnaire was pre-tested by partners of the BIO-PLASTICS EUROPE project.
[40]	Soares J	2021	To analyze perceptions about plastic pollution and its impacts as well as sociodemographic and psychological factors predicting individuals’ pro-environmental behaviors in the Portuguese context.	Portugal	Adults (aged 18 or older)	Not indicated	428/NA	The link to the questionnaire was disseminated by email and social/personal networks to achieve a wider and varied sample.	Self-administration (online)	Not indicated	50 questions, divided into 4 sections. A Likert-type scale, ranging from 1 (strongly disagree) to 5 (strongly agree), was used to assess the level of agreement with each item. Before the general administration of the questionnaire, a pre-test was conducted, to identify potential weaknesses of the questionnaire and appropriately formulate the questions in a clear and understandable way. The feedback from a sample of 5 subjects was discussed and considered for the final version of the questionnaire.
[27]	Charitou A	2021	Exploring knowledge and attitudes toward marine plastic pollution and the EU Single-Use Plastics Directive	Greece	The questionnaire was boosted via social media advertising without restrictions or focus to social media users with particular characteristics, only targeting profiles with Greek Internet Protocol addresses (IPs), in order to limit the bias of the sample. In addition, through the snowball method, participants were also asked to distribute it by their social media too.	No restriction	374/NA	Snowball methodology	Self-administration (online)	[47,48,49]	14 questions. Response collection: open-ended and multiple choice. structured questionnaire developed specifically for this study in Greek.
[34]	Forleo MB	2021	(i) to identify homogeneous segments of people according to the importance they attach to different sources and impacts of plastic litter; (ii) to understand if behavioral aspects and personal characteristics emerged for each cluster of people.	Italy	No specific criteria are indicated: respondents were invited to take part in the survey and asked to send a questionnaire to link their acquaintances, in order to reach a wider and varied sample	Not indicated	605/NA	Snowball methodology	Self-administration (online)	[50,51,52]	8 questions. Response collection: 4-point Likert scale; multiple choice. The selection of variables was inspired by several studies. Pilot testing by ten respondents was performed to check comprehension.
[37]	Menzel C	2021	To systematically investigate valence- and risk-related attitudes towards plastic packaging, plastic waste, and microplastic.	Germany	No specific criteria are indicated.	Not indicated	212/NA	Participants were recruited via university email lists, social media, and a flyer at university facilities.	Self-administration (online, single-category implicit association test (SC-IAT)	[53,54,55]	The survey first required participants to respond to stimuli (words and images), and then to answer 12 questions on 5-point Likert scales.
[41]	Thiele CJ	2021	What is the level of concern about microplastics in relation to other environmental issues? What are the reasons for concern about microplastics? How do people perceive the hazardousness of microplastics? And do concern levels and hazardousness perception differ between lay people and people academically or professionally versed in the topic?	Multiple countries	18 or older	Not indicated	1681/NA	Snowball methodology	Self-administration (online)	Not indicated	13 questions. Response collection: single and multiple choice, Likert scale. A survey was designed in English and translated into Spanish, German, Italian, French, Polish, Greek, Croatian, Japanese, Thai, Indonesian, Malay, Portuguese, Chinese, and Arabic. Back-translation via Google Translate was performed.
[25]	Barbir J	2021	To assess European citizens’ perspectives regarding their plastic consumption, and to evaluate their awareness of the direct and indirect effects of plastics on human health in order to influence current behavior trends.	25 European countries	No specific criteria are indicated. The survey was distributed through faculty and scientific mailing lists related to sustainability.	No restriction	1000/NA	The survey was distributed through faculty and scientific mailing lists related to sustainability.	Self-administration (online)	[56,57,58,59,60]	20 questions. Response collection: 5-point Likert scale; multiple choice. Pre-testing was carried out to adjust for conciseness and clarity.
[26]	Cammalleri V	2020	To assess the level of knowledge and awareness of medical students and residents in public health with regard to the theme “microplastics pollution”, in order to evaluate their competence in such a problem and to evidence possible needs for information, training. and updating the future leading figures in Public Health.	Italy	Undergraduate or postgraduate students attending Public Health university courses at the Sapienza University of Rome	Not indicated	151/NA	The research project was presented to the Presidents of the selected university degree courses, who organized the meetings with the students. The project was explained to all the students in the classroom.	Self-administration (mode unclear)	[61,62,63,64,65]	13 questions. Response collection: ordinal scale; multiple choice. The questionnaire was elaborated ad hoc on the basis of scientific evidence and of an educational project on microplastics, and validated before the beginning of the study.
[29]	Deng L	2020	This study investigated the public’s perceptions and attitudes towards microplastics in Shanghai and used an ordered regression model to explore the public’s willingness to reduce microplastics and its influencing factors.	China	No specific criteria are indicated. Respondents were recruited in parks, subway stations, shopping malls, and other public places in 4 administrative districts of Shanghai and through social media.	Not indicated	437/480 (91%)	We recruited respondents in parks, subway stations, shopping malls, and other public places in 4 administrative districts of Shanghai and through social media.	Face-to-face interview	Not indicated	23 questions. Response collection: single and multiple choice. 50 respondents were pre-tested to avoid possible misinterpretations.
[31]	Dilkes- Hoffman L	Sept., 2019	To identify whether the general public views plastics as a serious environmental issue. Secondary aims include to understand what factors influence attitudes toward plastics, and to explore whether those attitudes motivate any personal reduction in plastic use.	Australia	Sample representative of gender, age, and state for the Australian population	Not indicated	2518/3028 (83.2%)	The market research company selected respondents using the quota method, meaning that the sample selected was representative of gender, age, and state for the Australian population.	Self-administration (email)	[66,67,68]	10 questions. Response collection: Likert scale, multiple choice and open-ended. The survey was developed based on a variety of periodic environmental surveys, and refined through several rounds of prototyping within the authors’ research groups and selected members of the public.
[30]	Dilkes- Hoffman L	May, 2019	To understand current knowledge and perceptions regarding bioplastics								8 questions. Response collection: Likert scale, multiple choice and open-ended.The survey was developed based on a variety of periodic environmental surveys, and refined through several rounds of prototyping within the authors’ research groups and selected members of the public.

NA = Not Applicable.

## Data Availability

All data generated or analyzed during this study are included in this published article and its additional information files.

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
