# Peer review of "Surveys of Knowledge and Awareness of Plastic Pollution and Risk Reduction Behavior in the General Population: A Systematic Review"

_ijerph, 2025, doi:10.3390/ijerph22020177_

Round 1

Reviewer 1 Report

Comments and Suggestions for Authors

The authors searched Medline and EmBase for  surveys designed to measure in the general population the level of knowledge about different types of plastics, the risks associated with plastic pollution, and the awareness on actions to reduce them. The viewpoint is novel however, there are some writing issues need to be addressed before its publication.

Introduction part: the last paragraph belongs to funding contents should be deleted.

Some of the abbreviations used in the study should be carefully checked and revised. Such as `CROSS` is no needed to use as it is only used one time.

Figure 1 should be revised to include the whole workflow (processing or analysis content is better to be added).

The current Table 1 is meaningless, please revise it with details.

For Figure 2-3, there are two captions of each, please keep one for each figure. Please consider to make discussions associated with these two figures.

Please add more contents in the text on how the authors calculated the question scores as described in Figure 2.

Please add more contents linked with the objectives in the Conclusions part.

Acknowledgments part should be given by a different content with the funding source. Such as `the authors thank someone who provided helpful information or data for this work`.

Author Response

Reviewer comment

Response

Introduction part: the last paragraph belongs to funding contents should be deleted.

As suggested we have deleted most of the paragraph, as the information was redundant

Some of the abbreviations used in the study should be carefully checked and revised. Such as `CROSS` is no needed to use as it is only used one time.

We believe “CROSS” should be maintained, as it is the name that identifies a well-known tool. We have now defined the abbreviation (IQR)

Figure 1 should be revised to include the whole workflow (processing or analysis content is better to be added).

We understand that more information on processing and analysis would be interesting, however we have created the diagram in compliance with the PRISMA guidelines

https://pmc.ncbi.nlm.nih.gov/articles/PMC8005925/

using an online tool developed by PRISMA

https://www.betterevaluation.org/tools-resources/prisma-flow-diagram-generator

The current Table 1 is meaningless, please revise it with details

We are sorry for the inconvenience. We realize there was a problem with formatting, which we have now solved with the Editorial office. Please find the complete table in our revised manuscript

For Figure 2-3, there are two captions of each, please keep one for each figure.

We apologize for the inconvenience, again caused by technical issues. Please refer to the revised manuscript for the correct figures

Please consider to make discussions associated with these two figures

We agree. We have now discussed the shortcomings mentioning the instrument that assessed them, with reference to the appropriate figures

Please add more contents in the text on how the authors calculated the question scores as described in Figure 2.

We thank the reviewer for pointing out the lack of information on the method used. We have rewritten the paragraph explaining how scores were calculated, please see “Quality Evaluation Tool” in the Methods section

Please add more contents linked with the objectives in the Conclusions part.

We completely agree that the Conclusions somehow appeared detached from the objectives. We have added a final paragraph explaining our conclusions in the context of our aims and findings.

Acknowledgments part should be given by a different content with the funding source. Such as `the authors thank someone who provided helpful information or data for this work`.

We agree with the reviewer, however we kindly ask to leave this text in the Acknowledgement as phrased, as it is a requirement of the funder for all publications pertaining to this project

Reviewer 2 Report

Comments and Suggestions for Authors

The manuscript under review is a review article that aims to systematize the results of surveys on public knowledge of plastic types, plastic-related risks, and actions to minimize the potential harms of plastic. The aforementioned topics fall within the scope of the journal from the point of view of the interplay between human beings and their physical, mental and social environments, and their impact on health.

There are occasional typos in the text (line 34, 42, etc.)

The authors state that opinion surveys on the study of the effects of plastic are sparse, varying in methodology and handling of results. The reviewer searched peer-reviewed literature on this issue and agrees with the authors of the manuscript. In this regard, the formulation of the study objectives as to identify survey instruments, and to analyze the content of the questions and highlight shortcomings looks quite logical and relevant.

The approach chosen by the authors to analyze the surveys is described methodologically correct, and the authors' results are clearly presented in the manuscript.

Nevertheless, from the reviewer's point of view, it is not quite clear how exactly these results can be applied by other researchers? From my point of view, the answer to this question in the form of practical recommendations can be added to the Conclusions section .

Author Response

Reviewer comments

Response

There are occasional typos in the text (line 34, 42, etc.)

We apologize, and thank the reviewer for pointing out the typos, which we have now corrected

The approach chosen by the authors to analyze the surveys is described methodologically correct, and the authors' results are clearly presented in the manuscript. Nevertheless, from the reviewer's point of view, it is not quite clear how exactly these results can be applied by other researchers? From my point of view, the answer to this question in the form of practical recommendations can be added to the Conclusions section .

We agree that providing some practical recommendations would make this work more interesting and useful. We have therefore added a final paragraph to the Conclusions, providing some suggestions stemming from our findings

Reviewer 3 Report

Comments and Suggestions for Authors

I have reviewed the article titled "Surveys of knowledge and awareness of plastic pollution and risk reduction behavior in the general population: A systematic review of study methodologies" and raised some concerns which needs attention.

In the Abstract

1- Lines 16-18 and line 21 need to be rephrased to be clearer,

2- Abbreviations have to be detailed when first mentioned.

In the Introduction

1- Line 34, please correct the way references are cited in the text,

2- please check abbreviations in lines 67-71.

In Materials and Methods

1- Please number subtitles, that makes it easier to follow,

2- English revision is required.

In Results

1- Please check table 1, what does the reader learn from it, and what footer do you mean?

2- please try to improve the quality of figures 2 and 3, also the caption should be under the figures only,

In the Discussion

1- The paragraph starting at line 279 is listing the shortcomings without referring to the correct practices!

2- The article ignored to refer to the conclusions of the reviewed papers, which shall add value to the article and respond to part of its title "Surveys of knowledge and awareness of plastic pollution and risk reduction behavior in the general population",

In the Conclusions

1- The conclusion would have better written if indicating the important elements and considerations which should better attended to when designing an ideal survey, instead of showing only the limitations.

Author Response

Reviewer comment

Response

In the Abstract

1- Lines 16-18 and line 21 need to be rephrased to be clearer

We agree, and have rephrased the text, improving clarity and flow

In the Abstract

2- Abbreviations have to be detailed when first mentioned.

We agree, however the abbreviation (IQR) is no longer present after rephrasing the text (as described above)

In the Introduction

1- Line 34, please correct the way references are cited in the text

We thank the reviewer and apologize for the typo, which we have now corrected

In the Introduction

2- please check abbreviations in lines 67-71.

We have streamlined the text, eliminating excessive details which were reported elsewhere in the manuscript

In Materials and Methods

1- Please number subtitles, that makes it easier to follow.

We have numbered subtitles in the Materials and Methods section as requested.

In Materials and Methods

2- English revision is required.

We have edited the section to improve clarity and flow

In Results

1- Please check table 1, what does the reader learn from it, and what footer do you mean?

We apologize for the inconvenience. We realize there was a technical problem with editing, which we have now solved with the editorial office. The revised manuscript now contains the correct material

In Results

2- please try to improve the quality of figures 2 and 3, also the caption should be under the figures only,

As for the table, there was a technical issue with formatting which has now been solved. Please see the correct figures in the revised manuscript

In the Discussion

1- The paragraph starting at line 279 is listing the shortcomings without referring to the correct practices!

We have now discussed the shortcomings mentioning the instruments that assessed them, which contain indications on the correct practices.

In the Discussion

2- The article ignored to refer to the conclusions of the reviewed papers, which shall add value to the article and respond to part of its title "Surveys of knowledge and awareness of plastic pollution and risk reduction behavior in the general population",

We agree it would be interesting to examine the conclusions of reviewed papers, however the analysis of survey results was not the aim of this review, and therefore we did not undertake such analyses. This may indeed be an interesting topic for future studies.

In the Conclusions

1- The conclusion would have better written if indicating the important elements and considerations which should better attended to when designing an ideal survey, instead of showing only the limitations.

We thank the reviewer for this important observation. We have now enriched the Conclusion section with suggestions inspired by our findings, giving this work a more pragmatic approach.

Round 2

Reviewer 3 Report

Comments and Suggestions for Authors

I have no more comments